



# Small Ice Particles at Slightly Supercooled Temperatures in Tropical Maritime Convection

Gary Lloyd[1,2], Thomas Choularton[1], Keith Bower[1], Jonathan Crosier[1,2], Martin Gallagher[1], Michael
Flynn[1], James Dorsey[1,2], Dantong Liu[1], Jonathan W. Taylor[1], Oliver Schlenczek[3,4], Jacob Fugal[3,5],
Stephan Borrmann[3], Richard Cotton[6], Paul Field[6,7], Alan Blyth[2]

[1]Centre for Atmospheric Science, University of Manchester, Manchester, M13 9PL, UK
[2]National Centre for Atmospheric Science (NCAS), Leeds, LS2 9JT, UK
[3]Institute for Atmospheric Physics, Johannes Gutenberg University of Mainz, and Particle Chemistry Department, Max
Planck Institute for Chemistry, Mainz, Germany
[4]Max Planck Institute for Dynamics and Self-Organization, Göttingen, Germany
[5]SeeReal Technologies, Dresden, Germany
[6]Met Office, Exeter, UK
[7]School of Earth and Environment, University of Leeds, Leeds

*Correspondence to*: Gary Lloyd (gary.lloyd@manchester.ac.uk)





**Abstract.**

In this paper we show that the origin of the ice phase in tropical cumulus clouds over the sea may occur by primary ice nucleation of small crystals at temperatures just between 0 and -5°C. This was made possible through use of a holographic

instrument able to image cloud particles at very high resolution and small size (6 µm). The environment in which the observations were conducted was notable for the presence of desert dust advected over the ocean from the Sahara. However, there is no laboratory evidence to suggest that these dust particles can act as ice nuclei at temperatures warmer than about -10°C, the zone in which the first ice was observed in these clouds. The small ice particles were observed to grow rapidly by vapour diffusion, riming, and possibly through collisions with supercooled raindrops, causing these to freeze and potentially

shatter. This in turn leads to the further production of secondary ice in these clouds. Hence, although the numbers of primary ice particles are small, they are very effective in initiating the rapid glaciation of the cloud, altering the dynamics and precipitation production processes.

## 1. Introduction

The formation of the first ice particles in convective clouds is poorly understood, partly due to the difficulty in measuring small particles that are potentially outside the resolution capability of many cloud microphysics instruments. Even with measurements approaching the required resolution (likely a few 10s of microns), understanding the origin of the first ice in natural free floating clouds is complicated by competing microphysical processes such as secondary ice production and the effects of potential seeding from outside. Making measurements in rapidly evolving, dynamic and turbulent convective

clouds is particularly challenging, as is knowing the composition of aerosol present in the atmosphere and their ice nucleating properties. After ice is initiated in convective clouds, rapid glaciation often follows (Hallett et al., 1978; Hobbs and Rangno, 1985; Koenig, 1963; Lawson et al., 2015; Rangno and Hobbs, 1991), producing ice crystal concentrations several orders of magnitude higher than the number of predicted Ice Nucleating Particles (INPs) (DeMott et al., 2010). This rapid glaciation of convective clouds, coupled with inadequate measurement resolution is a major challenge to capturing first

ice particles in-situ. The origin of these first ice particles and the physico-chemical properties of the INPs that contribute to their formation are thus poorly understood.

The Ice in Clouds Experiment – Dust (ICE-D) took place around the Cape Verde Archipelago, with the aim of studying the influence of Saharan dust aerosol on the microphysics of developing convective clouds over the Atlantic Ocean. The location

is an ideal natural laboratory for conducting measurements of dust outflow from Africa and investigating its potential impacts on cloud microphysical properties. The Sahara is the largest desert in the world and the most significant source of mineral dust in the atmosphere (Tanaka and Chiba, 2006). Mineral dust emission is driven by factors that produce wind speed maxima at the surface, including the mixing of momentum towards the surface from the nocturnal low level jet



(NLLJ) at the top of the night time boundary layer, day time convection, (Fiedler et al., 2013) and cold pool outflow from afternoon convection (Marsham et al., 2013). The Saharan Atmospheric Boundary Layer (SABL) reaches as high as 6 km during the summer (Gamo, 1996). According to Marsham et al. (2013) the distribution of uplifted or Aeolian dust within the SABL has three typical scenarios. Uplifted dust with clear air above, dust concentrated in an elevated layer with clearer air

underneath and dust profiles that are well mixed down to the ground. When Saharan dust advects away from the Sahara and over the Atlantic ocean it rises over cooler moist air forming an elevated layer known as the Saharan Air Layer (SAL) (Karyampudi and Carlson, 1988). This process results in a strong inversion, with warm dry air anomalies around 850 hPa that increase the lifting condensation level (LCL) and Level of Free Convection (LFC) that often supresses deep convection (Wong and Dessler, 2005). Mineral dust such as that transported from the Sahara Desert is an atmospherically important INP

(Koop, 2013) that plays a crucial role in the formation of ice particles in clouds at temperatures below ~ -15 °C (Diehl, 2014)

In this paper we present measurements of an isolated, growing cumulus cloud. The first penetration was around the freezing level close to cloud top, and further penetrations were conducted with increasing altitude/decreasing temperature as the cloud grew. We describe the cloud particles observed and the thermodynamic environment that the convective cloud developed

within.

## 2. Measurements and Analysis

The Facility for Airborne Atmospheric Measurements (FAAM) British Aerospace-146 (BAe-146) aircraft was used during the ICE-D experiment to penetrate developing cumulus clouds around the Cape Verde Archipelago. The aircraft aimed to

make an initial penetration as the cloud top was just below the freezing level. Subsequent penetrations were then made with increasing altitude/decreasing temperature, a few hundred metres below cloud top, following the developing cloud upwards. The microphysical properties at slightly supercooled temperatures in updraft regions of the clouds were studied, to look for first ice particles. In-situ measurements of cloud microphysical and aerosol properties in each case were provided by a suite of instruments that included: a 2 Dimensional-Stereo (2DS) probe, providing 10 μm resolution shadow images of

hydrometeors over the size range $10 < d_p < 1280$ μm (SPEC Inc., Lawson et al., 2006); a Cloud Droplet Probe (CDP-100 Version-2, Droplet Measurement Technologies (DMT), Boulder, USA) (Lance et al., 2010) for measurement of the cloud droplet size distribution over the range $3 < d_p < 50$ μm; a Passive Cavity Aerosol Spectrometer Probe (PCASP-100X, DMT) measuring the aerosol size distribution over the particle size range $0.1 < d_p < 3$ μm (Cai et al., 2013). The physical properties of individual refractory absorbing particles were characterised using a Single Particle Soot Photometer (SP2) (DMT,

Boulder, USA) (Stephens et al. 2003). Water vapour measurements were made with the Water Vapour Sensing System II (WVSS-II, SpectraSensors) (Vance et al., 2015; Fleming and May, 2004), which uses a near-infrared tuneable diode laser absorption spectrometer to measure atmospheric water vapour. The range and accuracy given by the manufacturer is 50-60000 ±50 ppmv or ±5%, of the measurement, whichever is greatest, though the lower limit of the instrument is unclear (Vance et al. 2015). Data used in this analysis from core instruments on the aircraft included temperature, measured using



Rosemount/Goodrich type 102 temperature sensors (Stickney et al., 1994), and information about aircraft altitude, speed and position provided by the GPS-aided inertial navigation system.

The 2DS shadow imaging probe was used for geometric analysis of particle shape and size. From this information discrimination between spherical and irregular particles was possible for hydrometeors $> \sim 100$ μm using a circularity criterion (Crosier et al., 2011). The categories generated using information about particle shape were; Low Irregular (LI, shape factor between 1 and 1.2), indicating liquid droplets, or newly frozen liquid droplets that maintain a spherical shape; Medium Irregular (MI, shape factor between 1.2 and 1.4), for increasingly irregular particles, possibly indicative of ice; High Irregular (HI, shape factor $> 1.4$) indicating ice particles. The Optical Array Imaging Software (OASIS) package was used to analyse data from the 2DS in the above way. The software was developed by the National Centre for Atmospheric Science (NCAS) and DMT. Further discussion of this can be found in Crosier et al. (2011). The 2DS was fitted with Korolev anti-shatter tips (Korolev et al., 2011) to reduce particle shattering artefacts and examination of Inter-Arrival Time (IAT) histograms was also used to further identify and remove shattered particles, Field et al. (2006a).

Key to the analysis of cloud particles presented in this paper, was the use of a holographic imaging probe (HALOHolo) from the Institute for Atmospheric Physics at the University of Mainz and Max Planck Institute for Chemistry, Mainz. This probe has the ability to image small cloud particles at high resolution. The HALOHolo instruments is an improved development based on the principle design introduced by Spuler and Fugal (2011) and Fugal and Shaw (2009). It has an effective pixel size of 2.96 μm on a $6576 \times 4384$ detector giving a sample volume of approximately $19 \times 13 \times 155$ mm. The sample volume was reduced slightly to remove potential shattering artefacts close to the windows. The resulting sample volume is 35.4 cm$^3$. With a sample rate of approximately 6 frames per second and typical airspeed of 140 m s$^{-1}$ at the cloud penetration altitudes, a 3-D volume of cloud was provided every 23.3 metres with a total sample volume of 212.4 cm$^3$ s$^{-1}$. The retrievable size range from the 3D samples is $\sim 6$ μm to 5 mm, which is constrained by detectability for small particles and the one particle limit for large particles. Similar to the processing of the 2DS data, the distinction between frozen hydrometeors and droplets is done by shape with the limitations expressed in Korolev et al. (2017).

## 2.1 Temperature Measurements

In-situ temperature measurements during science flights were made using de-iced and non de-iced Rosemount sensors. In cloud these are subject to wetting, causing evaporative cooling that depresses the temperature measurement (Dearden et al., 2014). Lenschow and Pennell (1974) found the wetting of exposed sensors during cloud penetrations depressed temperatures by as much as 1 °C relative to protected sensors. Larger temperature anomalies $> 1$ °C from Rosemount sensors have also been reported (Heymsfield et al., 1979) and during ICE-D similar temperature decreases in cloud were observed. Due to these potential problems measured water vapour mixing ratio measurements were used to derive ambient temperature in cloud as described in Dearden et al. (2014). Of the hygrometers on the aircraft the WVSSII instrument operating on a flush



inlet (referred to as WVSSII-F) was used for this analysis. The reason for this is that the WVSSII-F is not susceptible to artefact when penetrating liquid clouds (Vance et al. 2015). The performance of the WVSSII-F was compared to a second WVSSII instrument sampling through a Rosemount inlet (WVSSII-R). For this comparison only data out of cloud (CDP LWC <0.01 g m$^{-3}$) was selected, as the WVSSII-R measurement is subject to contamination during cloud penetrations. We found excellent agreement between the two instruments with an $r^2$ of 0.99 (Fig. 1).

The vapour mixing ratio measured by the WVSSII-F, in the presence of liquid water, was assumed to be equal to the saturation mixing ratio. This value was converted to a saturation vapour pressure using information from aircraft pressure measurements and the temperature calculated using the Clausius–Clapeyron equation. The derived temperatures for each penetration were compared to the in-situ measurements in cloud and out of cloud before penetrations (Fig. 2). In cloud temperatures measured by the Rosemount sensor were seen to be very unreliable, often depressed by > 2 °C relative to the measurement in dry air prior to the penetration and > 5°C relative to the derived temperature. Quite striking was the finding that the derived temperature in cloud during one penetration with an updraft of 15 m s$^{-1}$ was, ~ 4.5 °C higher than the in-situ out of cloud temperature and ~ 7 °C higher than the in-situ in cloud wetted temperature. This was the warmest penetration during case 1 and in the strongest updraft. When calculating the saturated adiabatic curve for the measured atmospheric conditions (Fig. 3 green dashed line) the derived temperature was seen to lay on the parcel curve suggesting that, in principle, the high derived temperature was possible for a parcel lifted from the Level of Free Convection (LFC). Derived temperatures for other penetrations were closer to the environmental temperature measured by a nearby dropsonde.

## 3. Results

### 3.1 Observations of cloud microphysics in a growing cumulus turret

A cloud-dust interaction flight (Flight No. b926) took place on 14$^{th}$ August 2015. The aircraft departed from Praia International Airport, on the island of Santiago (14.9453° N, 23.4865° W) at 1423 UTC to investigate a line of convection ~ 200 km south of the island. Eight straight and level penetrations were made into growing convective clouds with increasing height/decreasing temperature. Video stills from six of these penetrations and associated in-situ out of cloud temperatures (prior to penetration) can be seen in Figure 4. P1 (Fig. 4a) was the penetration with the highest temperature (lowest altitude). The 10 second mean out of cloud temperature before the penetration was -5.3 °C, with a mean derived temperature in cloud of just below freezing (-0.2 °C) (Fig. 5a). When calculating the saturated adiabatic temperature from the LCL (Fig. 2), the derived temperature agrees well with the predicted temperature of a rising parcel of air (in-cloud) at that altitude level. We analysed the forward facing camera (FFC) video from the aircraft and with information about the field of view, speed of the aircraft and distance from the cloud we estimated the cloud top to be ~ 440 metres above the altitude of the cloud penetration. In a saturated environment this suggests that the temperature at cloud top may be ~ 2 °C lower than the environment measured by the instruments on the aircraft. Vertical wind data measurements during the penetration peaked at




+ 15 m s$^{-1}$ (Fig. 5b), the strongest updraft measured during this case. The uncertainty in temperature measurements in-cloud can be seen in Figure 5a. Immediately before the penetration the out of cloud temperature measured by the Rosemount temperature sensor is -5.3 °C. Soon after entering the cloud the problem of wetting and evaporative cooling depresses the reading by several degrees. The derived temperature is calculated for in cloud conditions (for cloud Liquid Water Content

(LWC) > 0.01 g m$^{-3}$), and this quickly rises to a consistent mean value of -0.2 °C. The lag in derived temperature is related to the response time of the WVSS-II (Vance et al. 2014). Peak cloud droplet concentrations, measured using the CDP were 43 cm$^{-3}$ (Fig. 5c) in P1 with peak LWCs of 0.6 g m$^{-3}$. Imagery from the HALOHolo showed the microphysics to be dominated by liquid water (the mean HALOHolo cloud droplet concentration was 31 cm$^{-3}$). Figure 6 shows images of liquid hydrometeors for penetration 1. Although dominated by liquid hydrometeors, holograms revealed (Fig. 5c, 7a) small ice

crystals, some less than 50 μm in size, in strong updrafts. The mean HALOHolo ice concentration for penetration 1 (2.8 km in length) were 6.6 L$^{-1}$, with 1 second peak values sometimes as high as ~ 30 L$^{-1}$ (fig. 5c). Although accurate determination of the in-cloud temperature is difficult, the implication is that small ice particles just slightly below freezing can be seen in this developing cumulus cloud. Size distributions (Fig. 8) for P1 and P3 show a small mode of ice crystals observed by the holography. It is not possible to see these small ice particles in the 2DS imagery at this stage due to its limited resolution (10

μm). After this initial penetration, imagery from later penetrations at lower temperatures showed increasing numbers of ice particles (characterised as medium irregular or highly irregular particles in the analysis of 2DS, or high irregular for HALOHolo). P4 (T-derived -6.5°C) contained many small ice particles together with bigger frozen drops (Fig. 7b) indicating secondary ice processes becoming active. Size distributions show the increasing size and numbers of ice particles measured by both the HALOHolo and the 2DS (Fig. 8). These penetrations at colder temperatures fell outside of suitable conditions for

the identification of any potential first ice particles in these clouds due to potential contamination from efficient secondary ice particle production mechanisms.

### 3.2 Aerosol Properties

Figure 9 shows 1 Hz particle concentration data, binned as a function of altitude (250 m bin resolution) for all available ICE-D cloud-aerosol interaction flights. The different concentrations of dust existed in air masses with different thermodynamic properties. Included are profiles of cloud droplet concentrations (measured by the CDP), aerosol concentrations measured by the (PCASP and Hematite sub-micron particle concentration derived from a single particle soot photometer (SP2). CDP data were selected for in-cloud periods only (CDP Liquid Water Content (LWC) > 0.01 g m$^{-3}$) and PCASP data were removed

from any in-cloud periods using the same threshold. The aerosol data were also carefully quality controlled for any anomalous data points that may have resulted from shattering of precipitation particles on the inlet, or any other interaction with cloud hydrometeors.



The measurements of aerosol particles show elevated concentrations on a number of days between about 2 and 6 km. This is likely to represent the Saharan Air Layer (SAL) and associated dust particles (e.g. Flight No. b929), Liu et al. (2017) showed that hematite content derived from the SP2 was correlated with this layer and seen as a good tracer for dust aerosol. Hematite content in the marine boundary layer was shown to not correlate well with enhanced concentrations of particles measured by

the PCASP where these were likely to be sea salt aerosol with no hematite contribution. There are large variations in the PCASP concentrations in the SAL, but when dust plumes are present concentrations can be several 100 cm$^{-3}$ compared with a few 10s cm$^{-3}$ outside of these plumes. When comparing these dust concentrations with droplet numbers from the CDP during cloud penetrations, we generally found much lower concentrations of droplets versus the concentrations of dust in the SAL dust plumes. Price et al. (2018) presented information about the characteristics of dust collected on filter samples

during the flight presented in this paper (Flight No. b926). They found a mode particle diameter of ∼ 10 µm, which is smaller than the ice particles observed by the HALOHolo.

**4. Discussion and Conclusions**

Measurements of a growing cumulus cloud in a strong updraft (+15 m s$^{-1}$) showed the presence of small (< 100 µm) ice crystals measured with a holographic instrument that are likely to have formed close to the freezing level in a liquid dominated updraft. Detailed analysis of the ambient temperature (Fig. 2, 5a) suggest that the in cloud temperature was very likely higher than the out of cloud temperature (-5.2 °C) measured in-situ by a Rosemount de-iced sensor before penetration. A derived temperature in cloud using water vapour mixing ratio measurements suggested a mean temperature of -0.2 °C with

an estimated error of ∼ ± 0.5 °C. The calculation of a derived temperature was carried out due to sensor wetting problems with the in-situ measurement from the Rosemount thermometer making it unreliable in cloud (Figs. 3, 5a). Despite the uncertainty in the derived temperature the true in-cloud temperature is thought to have been somewhere between the out of cloud ambient temperature and the derived temperature, but is very likely to be significantly higher than the temperature measured in clear air before penetration. The reasoning for this is that a saturated parcel of air rising in a strong updraft will

be warmer than its surroundings. Differences between out of cloud and in-cloud temperatures have been found during other measurement campaigns. Lawson et al. (1990) found temperature differences of ∼ 3.5 ° C between the out of cloud and in-cloud temperatures in convective clouds with the use of a radiometric thermometer.

Concentrations of ice particles measured by the HALOHolo were variable, with mean concentrations of 6.6 L$^{-1}$ and 1 second

peak values of ∼ 30 L$^{-1}$. Some of this variability is explained by the small sample volume and sampling statistics – Figure 5c has sampling error shaded for the HALOHolo ice time series. The sampling error is often a significant percentage of the measured concentration. Despite this, the measurements show concentrations of small ice particles that are higher than would be expected through primary ice nucleation alone when viewed in the context of only slightly supercooled temperatures (e.g. DeMott et al., 2010). Potential seeding of the cloud from above was looked for through analysis of



instrumentation before and after penetration, with no evidence of this process taking place. Recirculation of ice particles was also thought unlikely due to the strong updraft in close proximity to the freezing level. Dust particles that could be mistaken for ice particles were also ruled out due to the measured aerosol size distribution during this flight being smaller than the ice crystals observed by the holography. This conclusion is further supported by the morphology of the particles (Figs. 5c, 7),

with some having the appearance of liquid drops that had frozen. In other cases we observed high concentrations of ice in updraft regions, but they were more complex and we could not rule out some of the mechanisms of cloud seeding highlighted above.

Measurements of aerosol properties around Cape Verde show the presence of the SAL after it is lifted over the MBL. This

layer varied from case to case, but higher concentrations of aerosol particles were often measured in this region between about 4-6 km. Liu et al. (2017) found that hematite derived from the SP2 correlated well with this layer – and is a key constituent of mineral dust particles. The cloud droplet concentrations found in growing convective clouds were just a few 10s cm$^{-3}$, much lower than the aerosol concentrations in the SAL. Although this suggests the aerosol in this layer did not contribute significantly as Cloud Condensation Nuclei (CCN) the presence of a background concentration of dust particles

outside of the enhanced plumes and is confirmed by a hematite signal at most levels (Fig. 9). Therefore, it is possible that dust aerosol from the Sahara plays a role in the developing cumulus clouds around Cape Verde.

   The INPs contributing to the primary ice nucleation here are unknown. Laboratory studies suggest similar dust to be active at significantly lower temperatures than the aircraft penetration levels at which the first ice particles were seen, and for that reason the dust is unlikely to be the INP active in this case. However, biological material, including bacterial cells, internally

mixed with the dust particles may possibly provide ice active sites on dust particles that are more likely to be active at the slightly supercooled temperatures in this case e.g. Obata et al (1999).

Observations of the first ice particles that form in clouds are a significant measurement challenge. Here we have used holography and highly targeted cloud penetrations at the freezing level, following developing clouds upwards, in an attempt

to understand where the first ice particles in tropical maritime convective clouds form. The high resolution imagery from the holographic instrument has enabled us to observe small ice particles in the strong updraft of a growing cumulus cloud. Current understanding would suggest that the concentrations of ice particles at the slightly supercooled temperatures in these clouds should be limited. Concentrations peaking up to a few 10s L$^{-1}$ in the updraft we observed potentially highlight the difficulty in observing in-situ first primary ice particles – primary ice may be inherently associated with the production of

secondary ice particles that enhance the concentration. There are multiple mechanisms of secondary ice production (Field et al. 2017). The most studied mechanism of secondary ice production is through the rime-splintering process, known as the Hallett-Mossop process (Hallett and Mossop, 1974). In this case at lower temperatures there is strong evidence for this mechanism being involved in the glaciation of the cloud. Other production mechanisms do exist though and may take place simultaneously with the formation of first ice in these clouds. One such mechanism is the production of secondary ice


particles through droplet shattering during the freezing process (Leisner et al. 2014). The holographic images (Fig. 7) are consistent with this theory, where particle morphology is a mix of frozen drops and small irregular particles that would be produced during the shattering process together with larger rimed hyrdrometeors. In the absence of evidence supporting primary ice nucleation as a mechanism to produce the concentrations of ice we see in this case a combination of primary ice

formation and production secondary ice at the same time is suggested as the most likely explanation for the observations. From the measurements presented in this paper we find that:

1) Ice particles are present in concentrations of a few 10s L$^{-1}$ at derived temperatures in cloud between 0 °C and -2 °C.

2) The temperature of the cloud measured in this case was too high for dust to be active as an INP.

3) The concentrations of ice particles present exceed the estimated concentration of INPs that may be ice active at these temperatures, such as biological material.

4) Observations could be explained by: i) more efficient INP; ii) recycling of ice in the downwelling mantle of the convective cloud and iii) a secondary ice production process.

*Data availability.* The data presented in this paper is available through the Centre for Environmental Data Analysis (CEDA) via the following URL: http://catalogue.ceda.ac.uk/uuid/b0348ba21b784ae783201200213d02f4

*Author contributions.* All authors contributed to the research and preparation of the paper

*Acknowledgements.* Airborne data were obtained using the FAAM BAe-146 Atmospheric Research Aircraft, which was operated by Airtask and jointly funded by the UK Natural Environment Research Council (NERC). We acknowledge support from NERC under grant NE/M001954/1. The deployment of the HALOHolo was supported by the German Research Foundation (DFG) under grant SPP 1294, as well as by an Advanced Research Grant of the European Research Council (ERC), Project 321040 (EXCATRO), and internal funds of the Max-Planck Society.

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





5 **Figures**

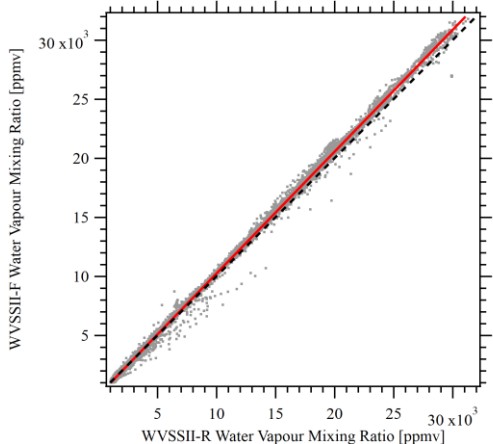

**Figure 1.** Comparison between WVSSII water vapour mixing ratio (ppmv) on the flush inlet and on the Rosemount inlet.

Red line represents least squares fit to the data ($r^2$ 0.99), black dashed line is the 1:1 line.



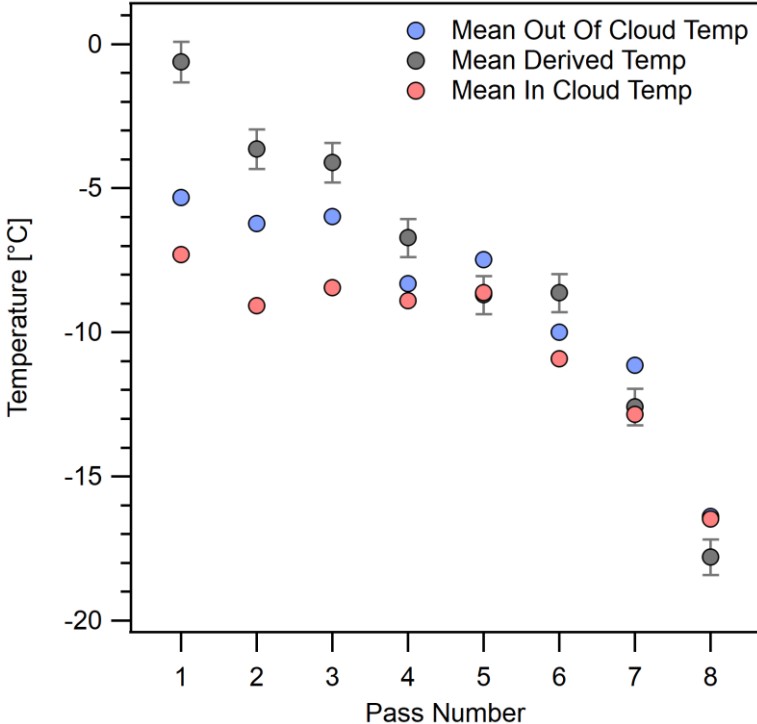

**Figure 2.** Temperatures as a function of penetration number from the Rosemount de-iced sensor out of cloud prior to penetration of cloud (blue markers), the wetted Rosemount de-iced sensor in cloud (red markers) and the derived temperature (grey markers). Error bars for the derived temperature represent propagation of ± 5% error in the WVSSII water vapour mixing ratio measurement.



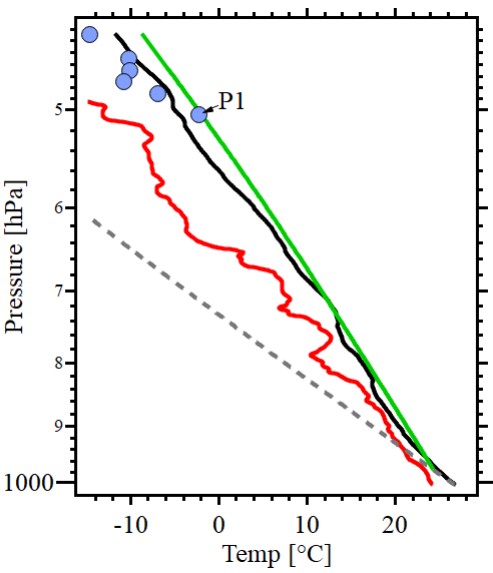

**Figure 3.** Pressure (hPa) vs Temperature (°C) during Case 1 obtained from: dropsonde measurements of temperature (black trace); dew point (red solid line); derived temperatures during cloud penetrations (blue symbols); calculated dry adiabatic lapse rate (grey solid line) and saturated adiabatic lapse rate (green solid line).


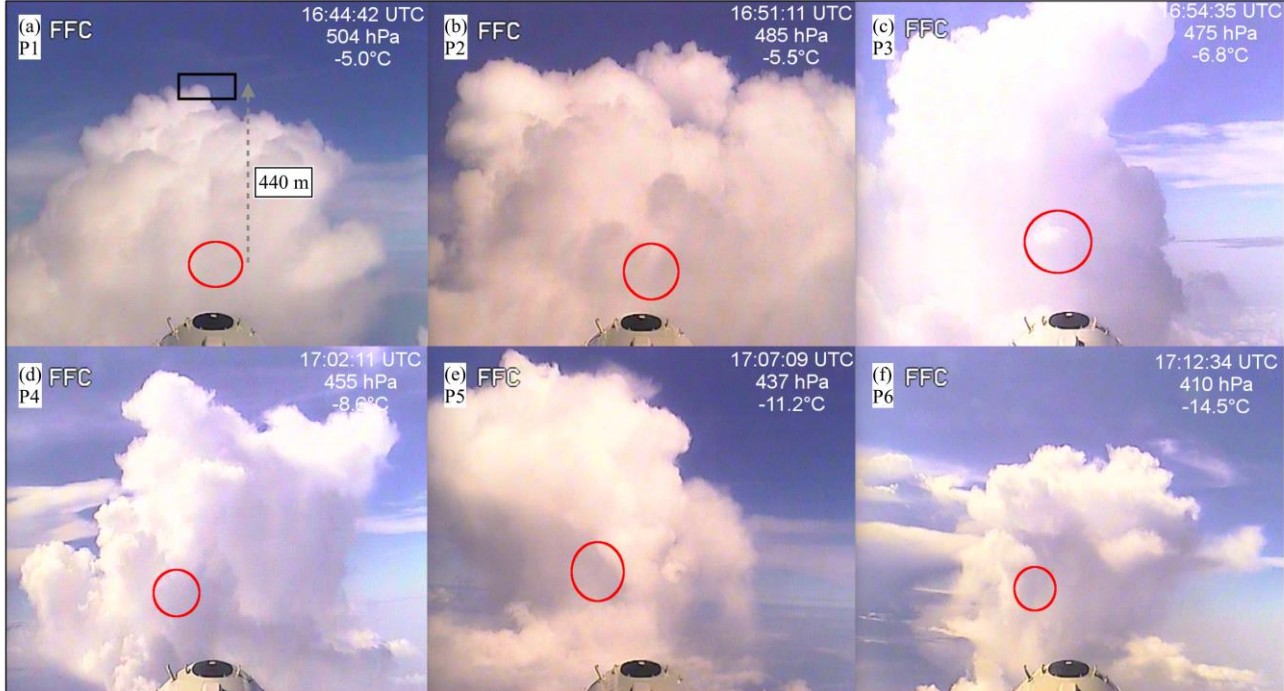

**Figure 4**. Images from the forward facing camera on the BAe-146 aircraft immediately before cloud penetrations (marked by red circles). Penetrations P1 to P6 are shown (from top left to bottom right). Temperatures are from the in-situ measurements out of cloud prior to each penetration. Image (a) P1 has the cloud top marked by a black box. The grey arrow marks distance to cloud top from penetration of 440 metres.



**Figure 5.** Time series from penetration 1. Axis (a): in-situ measured temperature from the Rosemount de-iced sensor (solid red line); derived temperature (dashed red line); mean derived temperature in cloud (grey marker) and mean in-situ measured temperature out of cloud before penetration (grey marker). Error bars show mean value integration period (x axis) and measurement error where applicable (y axis). Axis (b): vertical wind speed velocity (grey line) and axis (c): Liquid drop concentration from the HALOHolo (blue solid line); liquid drop concentration from the CDP (blue dashed line) and ice crystal concentration (multiplied by 1000 for comparison with liquid concentrations) from the HALOHolo (red solid line).





Superimposed imagery is included for ice crystals measured by the HALOHolo at time points 16:45:12 (i) and 16:45:15 (ii) and from the forward facing camera on the aircraft prior to penetration.

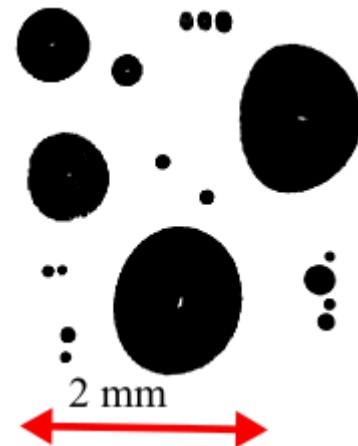

**Figure 6.** Holographic imagery indicating liquid droplets from Penetration 1

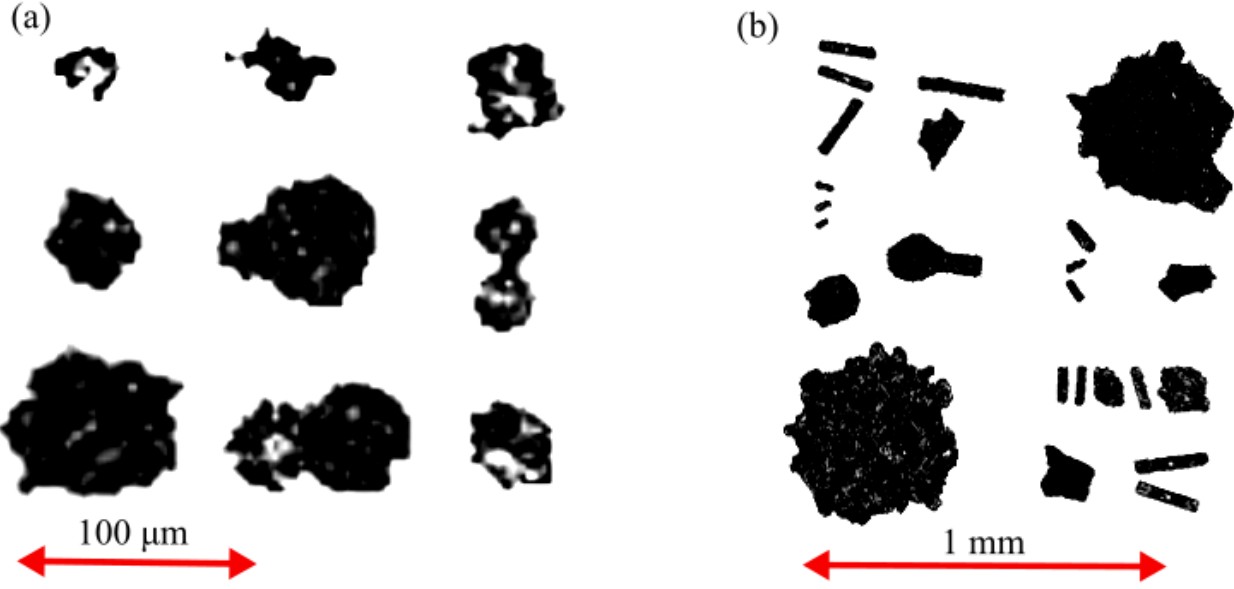

**Figure 7** Holographic imagery, indicating ice crystals, from Penetrations 1 and 4 (case 1).





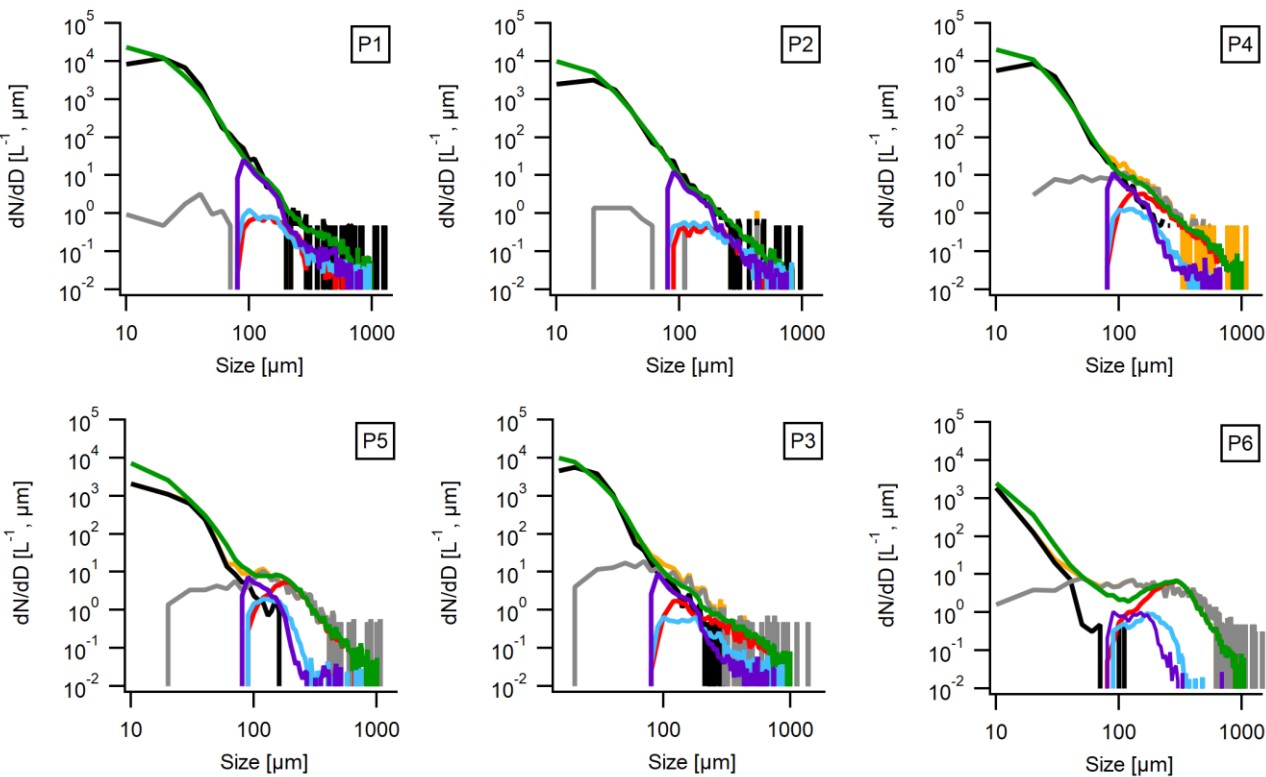

**Figure 8.** Particle Size distributions for 2DS All Particles (green line), Low Irregular (Purple Line), Medium Irregular (Light Blue Line), and High Irregular (Red Line) according to the characterised categories, and HALOHolo Spherical (Black Line) and Irregular Particles (Grey Line) categories.





**Figure 9.** Median concentration [cm$^{-3}$] as a function of altitude from the CDP (droplet concentrations, red circular symbols), aerosol concentrations from PCASP (blue circular symbols) and mean Hematite concentrations (green square symbols) from SP2. Error bars represent lower and upper quartile ranges. Flight numbers b919 – b933 are labelled on each plot.

