# Peer review of "Small Ice Particles at Slightly Supercooled Temperatures in Tropical Maritime Convection"

_Atmospheric Chemistry and Physics, 2019_

## Referee Comment (RC1) · Anonymous Referee #2 · 12 Jul 2019

General comment. Understanding the formation of ice in mixed phase cumulus clouds is hampered by our ability to measure small ice particles. Studies reported in this paper utilize a relatively recent technique for airborne ice measurements, using holography, which offers an improved method to sample small ice and to discriminate the signal from large cloud droplets. This paper represents an early example of how this technology can be applied to studying ice formation in cumulus clouds and will be of interest to the readers of this journal. However, the paper should be expanded and improved in some areas before it is published as discussed below.

In cloud temperature. A significant piece of this report involves in situ temperature measurements that appear to be as much as -7  $^{\circ}$ C lower than computed based upon the assumption of water saturation and measured water vapor concentrations. The

authors attribute this to evaporative cooling of the temperature sensor, a problem that has been reported before. However, the magnitude of this cooling appears to be higher than reported before and the analysis of this phenomenon is much less thorough than it needs to be to explain why this case should be different from previous studies. For example, evaporative cooling should not be constant across the cloud pass and temperatures and liquid water contents together should be compared with adiabatic values. Why was the wetting/cooling so much worse for the first three passes? The authors should consider, for example, some of the analysis techniques used by Lawson and Cooper (1990) to better explore and document their results. It would also be useful to compare their results with other studies on cloud buoyancy (e.g. some examples are given in Lawson and Cooper).

Secondary ice. The analysis of primary versus secondary ice formation is not thorough enough to be of much use. For example, is there sufficient evidence in their data, in terms of satisfying the criteria for rime-splintering (Hallett-Mossop) secondary ice mechanism, to support that as explaining (or not) the observed increases in ice concentration? The authors claim "in this case.... there is strong evidence for this mechanism being involved.." What strong evidence are they referring to? Is there enough information here to suggest the other mechanisms are active? How does the holographic data allow an improvement over previous studies? Etc.

Potential seeding from above. The paper claims "seeding of the cloud from above was looked for through analysis of instrumentation before and after the penetration, with no evidence of this process taking place." What analysis was done?

Dust and CCN. This section seems to be added on to the paper, without much tie-into the previous sections. The fact that there was dust present seems unremarkable given the location.

---

## Short Comment (SC1) · 5 Oct 2019

This is an interesting paper on an important topic and I have some suggestions for enhancing the paper. There are some published studies which should be discussed and provide some constraints on variables at the core of this paper:

1. At the core of this paper is a study of the first ice formation at relatively high super-cooled temperatures. On p 7 it is stated that "the measurements show concentrations of small ice particles that are higher than would be expected through primary ice nucleation alone when viewed in the context of only slightly supercooled temperatures (e.g. DeMott et al., 2010)." At this point a discussion of the INP measurements made from this aircraft during this campaign would be appropriate (Price et al., 2018). These

measurements were made from close to the surface up to several kilometres in air with a range of dust loadings and INP concentrations from the same flight (b926) were reported. While the technique only yielded results at lower temperature, the technique does define an upper limit at warmer temperatures of around 0.1 L-1. In addition, Price et al. also report estimates of marine organic INP and desert dust INP over the full temperature range from a global aerosol model.

Surface level INP measurements in Cape Verde have been reported in the literature (Welti et al., 2017). These measurements are probably representative of the marine boundary layer, hence are relevant for these convective systems. Welti never measured more than $\sim$ 1 m-3 at -5oC, again providing a constraint on primary ice production.

The fact that INP have not been observed at these warm temperatures in sufficient concentrations to produce the reported 6-30 ice crystals per litre suggests that some other factor is at play (as the authors go on to say, i.e. secondary production). The available measurements are not consistent with a population of bacterial INP as suggested on P8, ln 20.

2. The size distribution of dust is also extremely important for this paper. The authors suggest rough semi-spherical particles of 10s microns observed by HALOHolo are ice particles. This is partly justified on the basis that Price et al. "found a mode particle diameter of $\sim$ 10 $\mu$m, which is smaller than the ice particles observed by the HALOHolo". This can be improved. Ryder et al. (2018) present a study of the full size distributions measured from the FAAM aircraft, on the same deployment (ICE-D and AER-D were conducted at the same time). They report a substantial population of dust particles in the 10-100 $\mu$m size range. Are the concentrations of these rough spherical particles observed by HALOHolo really inconsistent with dust concentrations?

Technical comment

P3 ln 10. Koop 2013 is not in the reference list. If the authors are referring to the News and Views article, this probably isn't an appropriate citation.

References

Price, H. C., Baustian, K. J., McQuaid, J. B., Blyth, A., Bower, K. N., Choularton, T., Cotton, R. J., Cui, Z., Field, P. R., Gallagher, M., Hawker, R., Merrington, A., Miltenberger, A., III, R. R. N., Parker, S. T., Rosenberg, P. D., Taylor, J. W., Trembath, J., Vergara‐Temprado, J., Whale, T. F., Wilson, T. W., Young, G., and Murray, B. J.: Atmospheric Ice‐Nucleating Particles in the Dusty Tropical Atlantic, J. Geophys. Res., 123, 2175-2193, doi:10.1002/2017JD027560, 2018.

Ryder, C. L., Marenco, F., Brooke, J. K., Estelles, V., Cotton, R., Formenti, P., McQuaid, J. B., Price, H. C., Liu, D., Ausset, P., Rosenberg, P. D., Taylor, J. W., Choularton, T., Bower, K., Coe, H., Gallagher, M., Crosier, J., Lloyd, G., Highwood, E. J., and Murray, B. J.: Coarse-mode mineral dust size distributions, composition and optical properties from AER-D aircraft measurements over the tropical eastern Atlantic, Atmos. Chem. Phys., 18, 17225-17257, 10.5194/acp-18-17225-2018, 2018.

Welti, A., Müller, K., Fleming, Z. L., and Stratmann, F.: Concentration and variability of ice nuclei in the subtropic, maritime boundary layer, Atmos. Chem. Phys. Discuss., 2017, 1-18, 10.5194/acp-2017-783, 2017.
Interactive
comment

---

## Referee Comment (RC2) · Anonymous Referee #1 · 23 Oct 2019

The author present a case study of challenging measurements of cumulus clouds using a set of in-situ instruments, in particular a holographic imager. Surprisingly, ice crystal were found at temperature were the prevailing Sarah dust is not active in laboratory studies. The authors claim that secondary ice processes produced this the majority of these ice crystals The finding are relevant for publication and fits well in the scope of ACP. The authors describe well the measurements and discuss the results. I have only minor comments.

Page 2 Line2-10: Not sure if this detail discussion of the Sarah dust layer is necessary.

Page 4 Line 19: The resulting sample dimensions would be interested in addition or instead of the total sample dimensions.
Page 7 Line 10-11: Could a broad contribution with a mean mode diameter of 10 um have particles in the size range of the observed ice crystals?

Page 8 Line 2-4: Could the tail of a broad aerosol distribution be the source of the ice crystals? In particular as the aerosol concentration are a few magnitude higher than the ice crystals concentration.

Page 8 Line 4-5: I am not convinced that the pictures shows frozen droplets. The morphology would be better to judge from greyscale images and more example would be more convincing.

Figure 1 and 5c: Showing the exponent (x103) only on one tick can lead to confusion. I recommend using it on all ticks, showing it together with the unit [103 L-1] or using a different unit [cm-3].

Figure 6 and 7: Particular the small ice crystals in Figure 7a look larger than indicated by the scale. Can the authors please double check if the shown scale is correct? To get a better impression of the particles I recommend showing the greyscale images from the holographic imager.

**ACPD**

---

## Author Comment (AC1) · 1 Dec 2019

We thank the referees for their comments, which will make a valuable contribution to the quality of the manuscript. Please find our responses to the questions below.

**Anonymous Referee #2**

General comment. Understanding the formation of ice in mixed phase cumulus clouds is hampered by our ability to measure small ice particles. Studies reported in this paper utilize a relatively recent technique for airborne ice measurements, using holography, which offers an improved method to sample small ice and to discriminate the signal from large cloud droplets. This paper represents an early example of how this technology can be applied to studying ice formation in cumulus clouds and will be of interest to the readers of this journal. However, the paper should be expanded and improved in some areas before it is published as discussed below.

In cloud temperature. A significant piece of this report involves in situ temperature measurements that appear to be as much as -7  C lower than computed based upon the assumption of water saturation and measured water vapor concentrations. The authors attribute this to evaporative cooling of the temperature sensor, a problem that has been reported before. However, the magnitude of this cooling appears to be higher than reported before and the analysis of this phenomenon is much less thorough than it needs to be to explain why this case should be different from previous studies. For example, evaporative cooling should not be constant across the cloud pass and temperatures and liquid water contents together should be compared with adiabatic values. Why was the wetting/cooling so much worse for the first three passes? The authors should consider, for example, some of the analysis techniques used by Lawson and Cooper (1990) to better explore and document their results. It would also be useful to compare their results with other studies on cloud buoyancy (e.g. some examples are given in Lawson and Cooper).

**Author Response:** *In one pass we found a derived temperature that was 7C higher than the in cloud wetted reading from the rosemount temperature sensor. In other passes the difference was more modest (fig. 2 of the manuscript). We calculated our derived temperature using a water vapour sensor and this approach is described in section 2.1. of the manuscript.*

*The key measurements described are:*

- *The out of cloud rosemount forward facing temperature measurement*
- *The in cloud (wetted) rosemount forward facing temperature measurement*
- *The water vapour derived temperature measurement in cloud*

*Previous work by Lawson and Cooper (1990) used a reverse flow thermometer and compared it with a radiometer to examine the differences in reported temperatures. They found differences of about 3.5C between their out of cloud measurement with the reverse flow sensor vs the radiometer.*

*It is important to note the significant measurement differences in measurement technique, however we found all but one of our temperature differences during our penetrations to be within this range. The one penetration that fell outside this range had a 4.5C difference between the out of cloud measurement and the derived temperature. The 7C figure is the reported*

*difference between the in cloud wetted sensor and derived temperature. With regards to the derived temperature across the cloud pass we do find that it is variable and we calculated a mean derived value calculated for each cloud pass.*

*When we compare the derived temperature from pass 1 with the thermodynamic profile of the atmosphere (fig. 3 of the manuscript) we find it lies on the saturated adiabatic curve. Due to precipitation development we were sub-adiabatic with respect to LWC.*

Secondary ice. The analysis of primary versus secondary ice formation is not thorough enough to be of much use. For example, is there sufficient evidence in their data, in terms of satisfying the criteria for rime-splintering (Hallett-Mossop) secondary ice mechanism, to support that as explaining (or not) the observed increases in ice concentration? The authors claim "in this case. . .. there is strong evidence for this mechanism being involved.." What strong evidence are they referring to? Is there enough information here to suggest the other mechanisms are active? How does the holographic data allow an improvement over previous studies? Etc.

**Author Response:** *When considering the concentrations of ice we observed with the holographic instrument we found it difficult to explain them through primary ice nucleation alone as they were higher than would be expected given current understanding of the temperature dependent nature of heterogeneous ice nucleation. In response to the SC by Benjamin Murray we have added more detail (page 8 line 4-11) about INP measurements in the area.*

*Due to the enhanced concentrations of ice we considered a secondary ice production a possible source of ice. However we were careful not to draw any strong conclusions about the exact source of the ice, with a number of other potential mechanisms highlighted in conclusions (point 4).*

*This paper focuses on the evidence for first ice formation in the cloud studied, however we do briefly mention the development of the ice phase in subsequent passes where significant concentrations of ice were observed. The comment 'in this case…' relates to observations from colder clouds where graupel and columns were measured with the holography instrument. These findings are consistent with the process of rime-splintering, though we have re-phrased with sentence to highlight the uncertainty.*

Potential seeding from above. The paper claims "seeding of the cloud from above was looked for through analysis of instrumentation before and after the penetration, with no evidence of this process taking place." What analysis was done?

**Author Response:** *We looked at a range of in-situ instruments on the aircraft including the CDP and 2D-S to check for particle falling from above, or being present in the atmosphere. We found no evidence for this.*

Dust and CCN. This section seems to be added on to the paper, without much tie-into the previous sections. The fact that there was dust present seems unremarkable given the location.

**Author Response:** *We agree the presence of dust is unremarkable given the location but it could play a key role in the development of the cloud microphysics and the interpretation of our results.*

**Anonymous Referee #1**

The author present a case study of challenging measurements of cumulus clouds using a set of in-situ instruments, in particular a holographic imager. Surprisingly, ice crystal were found at temperature were the prevailing Sarah dust is not active in laboratory studies. The authors claim that secondary ice processes produced this the majority of these ice crystals The finding are relevant for publication and fits well in the scope of ACP. The authors describe well the measurements and discuss the results. I have only minor comments.

Page 2 Line2-10: Not sure if this detail discussion of the Sarah dust layer is necessary.

**Author Response:** *Due to the context of the campaign and focus on dust aerosol from the Sahara we would like to keep this section describing the transport of the aerosol in this region.*

Page 4 Line 19: The resulting sample dimensions would be interested in addition or instead of the total sample dimensions.

**Author Response:** *We have checked the manuscript and hope the dimensions you suggest are listed on page 4, lines 25*

Page 7 Line 10-11: Could a broad contribution with a mean mode diameter of 10 um have particles in the size range of the observed ice crystals?

**Author Response:** *We have considered this and the results are described in figure 1 and associated response in this document. We fine that we can't attribute our irregular particles to the dust measured during the flight.*

Page 8 Line 2-4: Could the tail of a broad aerosol distribution be the source of the ice crystals? In particular as the aerosol concentration are a few magnitude higher than the ice crystals concentration.

**Author Response:** *You are correct about the magnitude of the aerosol but the size range is smaller. This analysis is shown in figure 1.*

Page 8 Line 4-5: I am not convinced that the pictures shows frozen droplets. The morphology would be better to judge from greyscale images and more example would be more convincing. Figure 1 and 5c: Showing the exponent (x10^3) only on one tick can lead to confusion. I recommend using it on all ticks, showing it together with the unit [103 L-1] or using a different unit [cm-3].

**Author Response:** *We do feel that from studying the images they are likely to be frozen droplets, however as we are not certain we have added a sentence to highlight this. We have edited figure 1 and 5c to adjust the axis labelling.*

Figure 6 and 7: Particular the small ice crystals in Figure 7a look larger than indicated by the scale. Can the authors please double check if the shown scale is correct? To get a better impression of the particles I recommend showing the greyscale images from the holographic imager.

**Author Response:** *The images in figure 7a were blown up to make them easier to see, which is why they may look bigger. We checked the scale and it is correct – these are the small ice particles < ~ 100 um.*

**Benjamin Murray**
b.j.murray@leeds.ac.uk

This is an interesting paper on an important topic and I have some suggestions for enhancing the paper. There are some published studies which should be discussed and provide some constraints on variables at the core of this paper:

1. At the core of this paper is a study of the first ice formation at relatively high supercooled temperatures. On p 7 it is stated that "the measurements show concentrations of small ice particles that are higher than would be expected through primary ice nucleation alone when viewed in the context of only slightly supercooled temperatures (e.g. DeMott et al., 2010)." At this point a discussion of the INP measurements made from this aircraft during this campaign would be appropriate (Price et al., 2018).

These measurements were made from close to the surface up to several kilometres in air with a range of dust loadings and INP concentrations from the same flight (b926) were reported. While the technique only yielded results at lower temperature, the technique does define an upper limit at warmer temperatures of around 0.1 L-1. In addition, Price et al. also report estimates of marine organic INP and desert dust INP over the full temperature range from a global aerosol model.

Surface level INP measurements in Cape Verde have been reported in the literature (Welti et al., 2017). These measurements are probably representative of the marine boundary layer, hence are relevant for these convective systems. Welti never measured more than ~ 1 m-3 at -5oC, again providing a constraint on primary ice production.

**Author Response:** *We have added a discussion here of INP measurements. Welti et al. (2017) describe an upper temperature range of -5 C, which we describe in the manuscript. Interestingly the data presented in the paper figure 1 seems to suggests some activity as high as -3 C. (Page 8 lines 4-11)*

The fact that INP have not been observed at these warm temperatures in sufficient concentrations to produce the reported 6-30 ice crystals per litre suggests that some other factor is at play (as the authors go on to say, i.e. secondary production). The available measurements are not consistent with a population of bacterial INP as suggested on P8, ln 20.

**Author Response:** *We feel the sentence is very cautious with regards to the point made. We have added an extra sentence to clarify this.*

2. The size distribution of dust is also extremely important for this paper. The authors suggest rough semi-spherical particles of 10s microns observed by HALOHolo are ice particles. This is partly justified on the basis that Price et al. "found a mode particle di- ameter of ~ 10 μm, which is smaller than the ice particles observed by the HALOHolo". This can be improved. Ryder et al. (2018) present a study of the full size distributions measured from the FAAM aircraft, on the same deployment (ICE-D and AER-D were conducted at the same time). They

report a substantial population of dust particles in the 10-100 μm size range. Are the concentrations of these rough spherical particles observed by HALOHolo really inconsistent with dust concentrations?

**Author Response:** *The flight presented (b926) contained relatively low dust loadings in the context of the entire ICE-D/AER-D campaign. For that reason we would prefer to focus on the work by Price et al. (2018) which made measurements of the dust aerosol at ~2.4km during the same flight that we penetrated the growing cumulus cloud at 7.5 km. Fig. 1 below shows a comparison between the irregular particles measured by the holography and the dust measured during the same flight. The key point is that our ice size distribution from pass 1 (blue line) is 2-3 orders of magnitude above comparative measurements of the dust from the same flight.*

[Figure]

Technical comment

P3 ln 10. Koop 2013 is not in the reference list. If the authors are referring to the News and Views article, this probably isn't an appropriate citation.

**Author Response:** *Citation removed.*

References
Price, H. C., Baustian, K. J., McQuaid, J. B., Blyth, A., Bower, K. N., Choularton, T., Cotton, R. J., Cui, Z., Field, P. R., Gallagher, M., Hawker, R., Merrington, A., Mil- tenberger, A., III, R. R. N., Parker, S. T., Rosenberg, P. D., Taylor, J. W., Trembath, J., Vergara Rˇ Temprado, J., Whale, T. F., Wilson, T. W., Young, G., and Murray, B.

J.:AtmosphericIceâ˘Rˇ NucleatingParticlesintheDustyTropicalAtlantic,J.Geophys. Res., 123, 2175-2193, doi:10.1002/2017JD027560, 2018.

Ryder, C. L., Marenco, F., Brooke, J. K., Estelles, V., Cotton, R., Formenti, P., McQuaid, J. B., Price, H. C., Liu, D., Ausset, P., Rosenberg, P. D., Taylor, J. W., Choularton, T., Bower, K., Coe, H., Gallagher, M., Crosier, J., Lloyd, G., Highwood, E. J., and Murray, B. J.: Coarse-mode mineral dust size distributions, composition and optical properties from AER-D aircraft measurements over the tropical eastern Atlantic, Atmos. Chem. Phys., 18, 17225-17257, 10.5194/acp-18-17225-2018, 2018.

Welti, A., Müller, K., Fleming, Z. L., and Stratmann, F.: Concentration and variability of ice nuclei in the subtropic, maritime boundary layer, Atmos. Chem. Phys. Discuss., 2017, 1-18, 10.5194/acp-2017-783, 2017.